# Glaucoma Detection Using Support Vector Machine Method Based on Spectralis OCT

**DOI:** 10.3390/diagnostics12020391

**Published:** 2022-02-03

**Authors:** Chao-Wei Wu, Hsin-Yi Chen, Jui-Yu Chen, Ching-Hung Lee

**Affiliations:** 1Graduate Institute of Medicine, College of Medicine, Kaohsiung Medical University, Kaohsiung City 807378, Taiwan; chaowei196@gmail.com; 2Department of Ophthalmology, Kaohsiung Medical University Hospital, Kaohsiung Medical University, Kaohsiung City 807378, Taiwan; 3Department of Ophthalmology, Fu Jen Catholic University Hospital, New Taipei City 24352, Taiwan; 4School of Medicine, College of Medicine, Fu Jen Catholic University, New Taipei City 242062, Taiwan; 5Institute of Electrical and Control Engineering, National Yang Ming Chiao Tung University, Hsinchu City 30010, Taiwan; tp6bjo4m3r@gmail.com; 6Department of Electrical and Computer Engineering, National Yang Ming Chiao Tung University, Hsinchu City 30010, Taiwan

**Keywords:** optical coherence tomography (OCT), supported vector machine (SVM), glaucoma

## Abstract

Spectralis optical coherence tomography (OCT) provided more detailed parameters in the peripapillary and macular areas among the OCT machines, but it is not easy to understand the enormous information (114 features) generated from Spectralis OCT in glaucoma assessment. Machine learning methodology has been well-applied in glaucoma detection in recent years and has the ability to process a large amount of information at once. Here we aimed to analyze the diagnostic capability of Spectralis OCT parameters on glaucoma detection using Support Vector Machine (SVM) classification method in our population. Our results showed that applying all OCT features with the SVM method had good capability in the detection of glaucomatous eyes (area under curve (AUC) = 0.82), as well as discriminating normal eyes from early, moderate, or severe glaucomatous eyes (AUC = 0.78, 0.89, and 0.93, respectively). Apart from using all OCT features, the minimum rim width (MRW) may be good feature groups to discriminate early glaucomatous from normal eyes (AUC = 0.78). The combination of peripapillary and macular parameters, including MRW_temporal inferior (TI), MRW_global (G), ganglion cell layer (GCL)_outer temporal (T2), GCL_inner inferior (I1), peripapillary nerve fiber layer thickness (ppNFLT)_temporal superior (TS), and GCL_inner temporal (T1), provided better results (AUC = 0.84). This study showed promise in glaucoma management in the Taiwanese population. However, further validation study is needed to test the performance of our proposed model in the real world.

## 1. Introduction

Glaucoma is a progressive optic neuropathy, characterized by loss of retinal ganglion cells (RGCs) and their axons (the retinal nerve fiber layer [RNFL]), as well as the associated visual field (VF) defects [1,2,3]. Early glaucoma detection is crucial and important in managing this irreversible blinding disease [3,4]. It has been demonstrated that structural damage to the optic nerve head (ONH) and peripapillary RNFL (ppRNFL) can occur well before any detectable functional visual loss [5,6]. Spectral domain optical coherence tomography (SD OCT), which could measure the ONH and the ppRNFL, has been an important imaging modality in glaucoma practice [7,8,9]. One of the SD OCT instruments, the latest version of Spectralis OCT (Heidelberg Engineering, Inc., Heidelberg, Germany), Glaucoma Module Premium Edition (GMPE), could accurately determine the neuroretinal rim tissue by measuring the minimum distance between the Bruch membrane opening (BMO) and internal limiting membrane (ILM) [10,11]. A new parameter, the Bruch membrane opening–minimum rim width (BMO-MRW), has been shown to provide the most geometrically accurate measurement of the neuroretinal rim [10,11].

An emerging area of the diagnosis of diseases involves the use of automated interpretation of clinical data and digital images, with the help of artificial intelligence (AI) [12,13]. AI application in glaucoma detection and management has been well-discussed [13,14,15]. AI, coupled with OCT imaging, creates an algorithm that can be effectively used to make a model of complex data for detection, as well as diagnosis of glaucoma [12,16,17]. Machine learning methodology has been well-applied in glaucoma detection in recent years [16,17,18,19,20]. In our previous report, we developed automated classifiers to improve the discriminating power between glaucomatous and normal eyes with input parameters from Stratus OCT [18]. With the advancement of SD OCT technology, more detailed parameters derived from peripapillary and retinal areas could be provided. Here, we aim to analyze the diagnostic capability of Spectralis OCT on glaucoma detection using support vector machine (SVM) classification method in our population.

## 2. Materials and Methods

### 2.1. Participants

We enrolled healthy subjects, and primary glaucoma patients, meeting the eligibility criteria to this cross-sectional study. This research adhered to the tenets of the Declaration of Helsinki. Informed consent was obtained from all participants, and the study was approved by the Institutional Review Board of the Fu-Jen Catholic University Hospital (FJUH109021). Subjects with a best corrected visual acuity of less than 20/40, spherical equivalent outside −5.0 D, and cylinder correction of more than 3.0 D were excluded. To increase imaging quality and accuracy, patients with marked peripapillary atrophy were also excluded, in order to avoid instrumentation problems in the algorithms used to find the layers. All subjects underwent a complete ophthalmic examination, including slit lamp biomicroscopy, measurement of intraocular pressure (IOP), stereoscopic fundus examination, and standard full-threshold automated perimetry (30-2 mode, Humphrey Field Analyzer [HFA], model 750; Carl Zeiss Meditec, Inc., Dublin, CA, USA).

The patients with primary glaucoma, regardless open-angle or angle closure glaucoma, were recruited from a group of patients that had received at least 6 months of regular follow-up at the glaucoma service at the Fu-Jen Catholic University Hospital, between April 2019 and December 2020. Subjects with normal eyes were recruited from volunteers from the out-patient clinic and staff at the Fu-Jen Catholic University Hospital during the study period.

Eyes were defined as glaucomatous if there was both glaucomatous optic neuropathy (GON) and a reproducible glaucomatous visual field defect, in the absence of any other abnormalities to explain the defect. GON was defined as either inter-eye cup-disc ratio asymmetry >0.2, rim thinning or notching, peripapillary hemorrhages, or cup-disc ratio ≥0.6. Healthy eyes were defined as history of eye disease, no family history of glaucoma, IOP lower than 21 mm Hg, and normal optic disc appearance, based on clinical stereoscopic examination (no diffuse or focal rim thinning, optic disc hemorrhage, or RNFL defects) by the same experienced doctor (H.Y.C, glaucoma specialist). A normal result on the Glaucoma Hemifield Test and corrected pattern SD (HFA, program 30-2), within normal limits, were required.

In total, 498 glaucomatous eyes (mean deviation: −6.09 ± 7.16 dB) and 254 normal eyes (mean deviation: −0.80 ± 1.31 dB) were studied.

### 2.2. Visual Field Testing

Achromatic automated perimetry was performed with an HFA, with the central full-threshold visual field-testing program 30-2. Visual field reliability criteria included fixation losses and false-positive and -negative rates of less than 20%. The evaluation of glaucomatous visual field defects was made based on the following liberal criteria: two or more contiguous points with a pattern deviation sensitivity loss of *p* < 0.01; three or more contiguous points with sensitivity loss of *p* < 0.05, in the superior or inferior arcuate areas; a 10-dB difference across the nasal horizontal midline at two or more adjacent locations; and an abnormal result on the glaucoma hemifield test [21]. Glaucoma severity was staged by Hodapp, Parish and Anderson criteria [22]. The visual filed index including mean deviation (MD) and pattern standard deviation (PSD) were used for analysis.

### 2.3. Spectralis OCT (Heidelberg Engineering GmbH) Imaging

All participants were examined using the optic nerve head radial and circular (ONH-RC) and posterior pole horizontal (PPoleH) scan protocols, implemented in the new Glaucoma Premium Module Edition (GPME) by the Spectralis OCT device.

The ONH-RC scan protocol comprised of 24 equally spaced radial B-scans, each with 768 A-scans, covering a 15° region, centered on the optic disc, for measurement of BMO-MRW, as well as a 3.5 mm diameter circle scan for measurement of peripapillary retinal nerve fiber layer (ppRNFL) thickness. Twenty-five B-scans were captured and automatically averaged for each B-scan location. The ppRNFL and BMO-MRW measurements are displayed in seven parts, the temporal (T), temporal inferior (TI), nasal inferior (NI), nasal (N), nasal superior (NS), temporal superior (TS), and global (G) areas.

The PPoleH scan protocol consists of 61 horizontal B-scans, centered on the fovea, oriented to the fovea-disc axis, and symmetrically distributed in the upper and lower hemispheres. It provided full-layer retinal thickness maps and automatic segmented thickness maps for each retinal layer, which are displayed in two modes, the: 1, 3, 6 mm early treatment diabetic retinopathy study (ETDRS) grid and 8 × 8 grid.

In the 1, 3, 6 mm ETDRS grid mode, the full-layer retinal average thickness and average thickness of each retinal layer are provided in nine subfields, defined by ETDRS. The diameters of the inner, intermediate, and outer rings are 1, 3, and 6 mm, respectively. The average of all points within the inner ring area is defined as the central thickness (C). The intermediate ring is divided into four sectors, the inner temporal (T1), inner inferior (I1), inner nasal (N1), and inner superior (S1) sectors. The outer ring was divided in the same fashion, named the outer temporal (T2), outer inferior (I2), outer nasal (N2), and outer superior (S2) sectors. Only the full-layer retinal average thickness and average thickness of nerve fiber layer, ganglion cell layer, and inner plexiform layer in the 9 subfields were included in this study.

In the 8 × 8 grid mode, the thickness of the entire retina of the central 24° area of posterior pole is measured, averaged, and displayed in an 8 × 8 grid. We labeled the 64 measurements of the full-layer retinal thickness, with the first number representing the order from top to bottom and second number representing the order from temporal to nasal site.

The labels of the location of each parameter group were shown in Figure 1. All scans were acquired, with reference to the subject’s specific fovea-BMO (FoBMO) axis. Images had to have a quality index of at least 20 to be included in the study. Images with artifacts were excluded.

### 2.4. Performance of the Overall Feature Groups

We divided the parameters into nine feature groups (Table 1): AGR (age, gender, refraction; 3 features), minimum rim width (MRW: T, TI, NI, N, NS, TS, G; 7 features), peripapillary nerve fiber layer thickness (ppNFLT: T, TI, NI, N, NS, TS, G; 7 features), retinal average thickness (RAT: T1, T2, I1, I2, N1, N2, S1, S2, C in 1, 3, 6 mm ETDRS grid; 9 features), nerve fiber layer (NFL: T1, T2, I1, I2, N1, N2, S1, S2, C in 1, 3, 6 mm ETDRS grid; 9 features), ganglion cell layer (GCL: T1, T2, I1, I2, N1, N2, S1, S2, C in 1, 3, 6 mm ETDRS grid; 9 features), inner plexiform layer (IPL: T1, T2, I1, I2, N1, N2, S1, S2, C in 1, 3, 6 mm ETDRS grid; 9 features), retinal average thickness in 8 × 8 grid (RAT 8 × 8; 64 features), and OCT (all OCT features; 114 features). The demographic and OCT features of the normal and glaucoma groups were compared using two-sided *t*-test in the statistical package of SciPy [23]. We experimented with fitting each of the nine feature groups, in order to discriminate glaucomatous eyes from the normal eyes and further recognize the glaucomatous eyes in different stages. All classifiers was trained using the SVM method [24], as implemented in libsvm [25], with 10-fold cross-validation; the kernel function is the radial basis function. The performance was evaluated by sensitivity, specificity, accuracy, and area under curve (AUC). All the data processing and model training algorithms were developed with Pandas [26] and Scikit-learn [27].

### 2.5. Feature Selection with Mutual Information

The mutual information (MI) between random variables X and Y is defined as follows:(1)$$MI\left(X;Y\right)=D_{KL}\left(p\left(X,Y\right)∥p\left(X\right)p\left(Y\right)\right)=\underset{y\in Y}{\sum }\underset{x\in X}{\sum }p\left(x,y\right)\log\frac{p\left(x,y\right)}{p\left(x\right)p\left(y\right)}$$
where *D_KL_* is the Kullback–Leibler divergence, $p\left(X,Y\right)$ is the joint probability mass function of *X* and *Y*, and $p\left(X\right)$ and $p\left(Y\right)$ are the marginal probability mass function of *X* and *Y*, respectively.

The $MI\left(X;Y\right)$ measures the dependence of two random variables, in terms of the similarity of their distributions. By the non-negative property of Kullback–Leibler divergence, $MI\left(X;Y\right)$ is also non-negative with the lower bound $MI\left(X;Y\right)=0$, if and only if $p\left(x,y\right)=p\left(x\right)p\left(y\right)$; that is, *X* and *Y* are independent random variables [28]. A larger than zero MI means that knowing *X* give a certain extent of deterministic value of *Y*. The higher the MI is, more information is shared between the two random variables.

We calculated MI using each of the 114 OCT features and 3 clinical features (age, gender, and refraction), as X and glaucomatous were positive or not as Y. Features were iteratively selected from the top 20 pool, ranked by MI and trained with SVM to obtain the average performance of 10-fold cross-validation. Only the best feature was kept in the final subset for each iteration. The procedure repeated by adding new features and comparing the performance until the subset contained 10 selected features, which is the amount generally easier to interpret by human. Only those subsets that had test results which improved from the previous iteration were kept for further discussion.

## 3. Results

### 3.1. Demographic and Clinical Data

The demographic and clinical characteristics of the study groups are presented in Table 2. The mean age was 52.50 ± 16.19 years in the normal group and 59.20 ±13.03 years in the glaucoma group. There was significant difference in age between the two groups (*p* < 0.001). Visual field parameters, including MD and PSD, showed significant differences (*p* < 0.001). No significant difference was observed in the refraction (*p* = 0.104). Regarding the OCT parameters, all parameters were significantly different between the two groups, except NFL_N1.

Table 3 shows the amounts of features in each feature group and SVM classification results in differentiating normal from all glaucomatous eyes, based on each feature group. The OCT feature group yielded the best AUC value (AUC = 0.82), followed by MRW (AUC = 0.81) and ppNFLT (AUC = 0.81). Figure 2 showed the ROC curves of all the feature groups in differentiating normal from all glaucomatous eyes.

Table 4 reveals the SVM classification results, using AGR feature group solely, and the above OCT-related feature groups (plus AGR) to differentiate normal from all glaucomatous eyes. The AUC value of using AGR feature group solely for classification is 0.55. The OCT + AGR and ppNFLT + AGR feature group had the highest AUC value (AUC = 0.82). The ROC curves of all the OCT-related feature groups (plus age, gender, and refraction), in differentiating normal from all glaucomatous eyes, was shown in Appendix A.

Table 5 shows the SVM classification results, between normal and different stages of glaucomatous eyes, using Spectralis OCT feature groups. The OCT feature group had the best performance in discriminating between normal and early, moderate, or severe glaucomatous eyes, with AUC values of 0.78, 0.89, and 0.93, respectively. The MRW feature group also had good results in distinguishing normal from early glaucomatous eyes, with an AUC value of 0.89, and the RAT 8 × 8 feature group also had good performance in discriminating normal from severe glaucomatous eyes, with an AUC value of 0.93. The ROC curves of all OCT-related features groups, in differentiating normal from varying stages of glaucomatous eyes, were shown in Figure 3.

Table 6 demonstrates that the SVM classification result between normal and different stage of glaucomatous eyes using Spectralis OCT feature groups plus age, gender, and refraction. The OCT + AGR feature group had the highest value of AUC in discriminating between normal and early, moderate, and severe glaucomatous eyes with values of 0.78, 0.89, and 0.93, respectively. The ppRNFL + AGR feature group also had good result in distinguishing normal from moderate glaucomatous eyes with an AUC value of 0.89, and NFL + AGR feature group also had good performance in discriminating normal from severe glaucomatous eyes with an AUC value of 0.93. Appendix A showed the ROC curves of all the OCT-related feature groups, plus age, gender, and refraction, in differentiating normal from varying stages of glaucomatous eyes.

### 3.2. Selected Features and Generalized Detecting Model

Table 7 shows six subsets of features after the selecting procedure described in the previous section. We named these subsets of mutual information (MI) 1, MI 2, MI 4, MI 6, MI 8, and MI 10 for the best result in iterations of one, two, four, six, eight, and ten features, respectively. The top 10 features are MRW_TI, MRW_G, GCL_T2, GCL_I1, ppNFLT_TS, GCL_T1, ppNFLT_TI, IPL_T2, MRW_TS, and ppNFLT_G.

The SVM classification results using above selected featured subsets were listed in Table 8. The MI 6, a combination of MRW_TI, MRW_G, GCL_T2, GCL_I1, ppNFLT_TS, and GCL_T1, outperformed the other subsets and obtained an average 0.84 of AUC for cross-validation. The MI 8 and MI 10 subsets did not have improved performance over the MI 6 after adding new features. Figure 4 showed the comparison of ROC curves and the corresponding AUC values of the 6 selected feature subsets in differentiating normal from all glaucomatous eyes.

## 4. Discussion

Spectralis OCT provided more detailed parameters in peripapillary area and macular areas among the OCT machines [9]. In the real glaucoma practice, however, it is not easy to integrate the enormous information (114 features), generated from Spectralis OCT, in glaucoma management. Machine learning classifiers, on the other hand, are proven analytical methods, especially good at detecting relationships between large numbers of input parameters, producing reliable classification results [29]. Therefore, if the machine learning method is combined with the parameters of Spectralis OCT, it may provide an effective and efficient assessment for the diagnosis of glaucoma.

SVM is a supervised machine learning classifier, which is one of the most powerful and robust classifications, widely used to deal with binary classification problems in various fields [24,30,31,32,33]. It has also been used for glaucoma detection in previous studies and provided promising results [19,34,35,36]. Compared with other machine learning approaches, SVM maps the nonlinearly separable data into a high-dimensional space through kernel functions, in order to transfer the corresponding to a linearly separable state. It maintains high generalization ability of the learning machine simultaneously. Thus, SVM is relatively effective when solving problems with the number of feature dimensions greater than the number of samples [31], as in this study, we used abundant OCT parameters for glaucoma discrimination. In addition, for small data problems like ours, SVM still performs well in accuracy and is relatively memory efficient [24,33].

Some important and meaningful information were obtained from our results. First, it showed good capability using all Spectralis OCT parameters with SVM method in detection of glaucomatous eye (AUC = 0.82). Simply using the values of MRW or ppNFLT with SVM method may also have good performance (AUC = 0.81) in discrimination normal from glaucomatous eyes. Furthermore, SVM method based on all Spectralis OCT parameters not only showed good capability in detection moderate and severe glaucoma (AUC = 0.89 and 0.93, respectively), but also had acceptable performance in distinguishing early glaucomatous eyes (AUC = 0.78). The performance was similar when using the MRW parameters to detect early glaucomatous eyes (AUC = 0.78).

Because there were significant differences in some demographic characteristics between our normal and glaucoma groups, we added age, gender, and refraction information into training the SVM model to show that the classification performance is mostly directly due to the OCT features but not due to the demographic differences between the two groups. The results showed that for discriminating normal from glaucomatous eyes, adding age, gender and refraction information did not change the value of AUC drastically, and using all OCT features was still the best one with the same AUC value (AUC = 0.82). For distinguishing different stages of glaucomatous eyes from the normal eyes, adding age, gender and refraction information still did not influence the performance of classification much in most of the feature groups. Lastly, the technique to use MI as the selecting index could intuitively select a combination of various best features from each group to complement each other. The best subgroup (MI 6) contains only six features (MRW_TI, MRW_G, GCL_T2, GCL_I1, ppNFLT_TS, and GCL_T1), generated from 114 OCT features, as well as three clinical features (age, gender, and refraction), provided a good predicting model, as our result showed an AUC of 0.84. Further validation studies, with more cases, are needed to test the performance of our proposed model in the real world.

To our knowledge, our study was the few ones which evaluated the application of machine learning technique in complicated Spectralis OCT parameters for glaucoma detection, including ppRNFL, ONH, and macular parameters. Several published literatures have explored the use of Spectralis OCT parameters to construct machine learning classifiers for glaucoma diagnosis [16,37,38,39]. Kim et al. developed several machine learning models, including SVM for glaucoma diagnosis, using ppRNFL parameters and clinical features (age, IOP, and corneal thickness) and visual field information, and they found the random forest model had the best performance, with an AUC value of 0.979 and AUC value of the SVM model at 0.967 [37]. Oh et al. also constructed several machine learning models, including a SVM using three ppRNFL measurements (ppRNFL superior, ppRNFL inferior, and ppRNFL temporal), as well as IOP and PSD for glaucoma detection, and the extreme gradient boosting model was shown to be the best model, with an AUC value of 0.945, the same AUC value as the SVM’s but with higher accuracy, sensitivity, and specificity. ppRNFL superior, ppRNFL inferior, and PSD were found to have a stronger influence in their proposed prediction model [16]. Park et al. used a multilayer neural network to combine BMO-MRW and ppRNFL parameters for glaucoma diagnosis, which showed better performance than using either BMO-MRW or ppRNFL data alone [38]. A deep learning classification model was adopted by Seo et al. for discriminating early normal tension glaucoma from glaucoma, which suspected and showed the best performance, considering three OCT-based parameters together (BMO-MRW, ppRNFL, and the color classification of ppRNFL), with an AUC value of 0.966 [39]. Though it is difficult to directly compare our results with previous research, due to the differences in the subjects included, as well as the OCT parameters and machine learning methods used. The above papers and ours had proved that it is feasible to construct reliable machine learning classifiers using Spectralis OCT parameters for glaucoma diagnosis. Unlike previous studies, our study not only used the ONH and ppRNFL parameters but also covered macula-related parameters, in order to have a more comprehensive analysis.

Although our results are interesting and promising, there are some limitations in our study. First, the substrate for studies is usually a clinic-based population of patients with glaucoma. These patients have been identified on the basis of particular patterns of structural and functional abnormality that meet preconceived notions that bias the outcome of the comparisons [40]. Therefore, this could overestimate the diagnostic accuracy of OCT instruments, which is a common problem in this type of case–control study. Furthermore, in our study subjects, there is significant difference in age between the glaucoma group and normal group. As found in previous studies [41], age may have an effect on the OCT measurements of the peripapillary retinal nerve fiber layer, macula, and optic head, which may also be a limitation of this study. Another limitation is the relatively small samples used to generate this model. Larger sample sizes are recommended to provide more precise and robust estimations for glaucoma diagnosis using machine learning methods.

## 5. Conclusions

SVM application to Spectralis OCT shows good diagnostic capability in differentiating glaucomatous from normal eyes. Our results show promise in glaucoma management in the Taiwanese population. However, the OCT result should be incorporated with other clinical information before decision-making. Further validation studies are needed to test the performance of our proposed model in the real world.

## Figures and Tables

**Figure 1 diagnostics-12-00391-f001:**
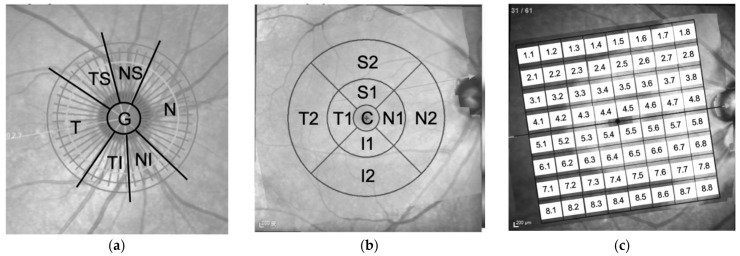
The labels of the location of (**a**) peripapillary retinal nerve fiber layer (ppRNFL) and Bruch membrane opening–minimum rim width (BMO-MRW) parameters, (**b**) retinal average thickness and thickness of each retinal layer in ETDRS grid, and (**c**) retinal average thickness in 8 × 8 grid. temporal (T), temporal inferior (TI), nasal inferior (NI), nasal (N), nasal superior (NS), temporal superior (TS), global (G), Superior (S), Inferior (I), Central (C).

**Figure 2 diagnostics-12-00391-f002:**
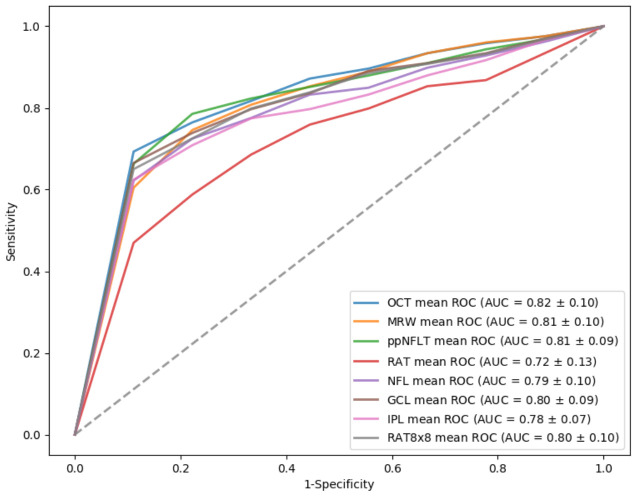
Receiver operating characteristic (ROC) curve of optical coherence tomography (OCT)-related features groups in differentiating normal from glaucomatous eyes.

**Figure 3 diagnostics-12-00391-f003:**
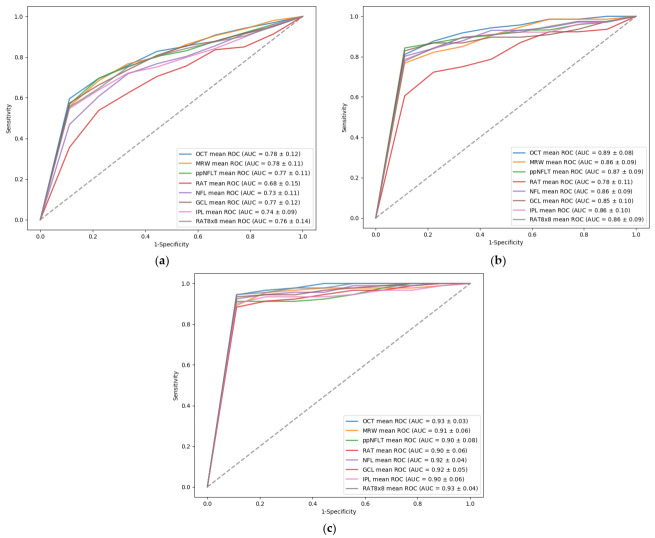
ROC curve of OCT-related features groups in differentiating normal from (**a**) early, (**b**) moderate, and (**c**) severe glaucomatous eyes.

**Figure 4 diagnostics-12-00391-f004:**
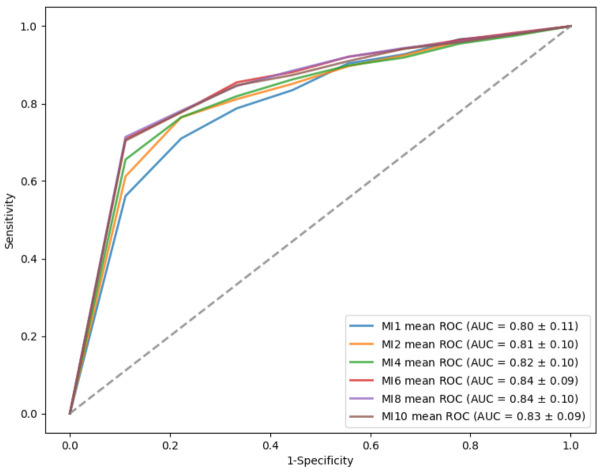
ROC curve of six selected feature subsets in differentiating normal from glaucomatous eyes.

**Table 1 diagnostics-12-00391-t001:** Feature groups.

Groups	Amounts of Features	Features
AGR	3	Age, gender, refraction
MRW	7	Minimum rim width: T, TI, NI, N, NS, TS, G
ppNFLT	7	Retinal nerve fiber layer thickness: T, TI, NI, N, NS, TS, G
RAT	9	Retina average thickness in 1, 3, 6 mm ETDRS grid: T1, T2, I1, I2, N1, N2, S1, S2, C
NFL	9	Nerve fiber layer in 1, 3, 6 mm ETDRS grid: T1, T2, I1, I2, N1, N2, S1, S2, C
GCL	9	Ganglion cell layer in 1, 3, 6 mm ETDRS grid: T1, T2, I1, I2, N1, N2, S1, S2, C
IPL	9	Inner plexiform layer in 1, 3, 6 mm ETDRS grid: T1, T2, I1, I2, N1, N2, S1, S2, C
RAT 8 × 8	64	Retina average thickness in an 8 × 8 grid
OCT	114	All of the above features except age, gender and refraction

T: temporal; TI: temporal inferior; TS: temporal superior; N: nasal; NI: nasal inferior; NS: nasal superior; G: global; T1: inner temporal; T2: outer temporal; I1: inner inferior; I2: outer inferior; N1: inner nasal; N2: outer nasal; S1: inner superior; S2: outer superior; C: central.

**Table 2 diagnostics-12-00391-t002:** The demographic and clinical characteristics of the study groups.

	Normal (*n* = 254 Eyes)	Glaucoma (*n* = 498 Eyes)	*p*-Value *
Gender			
Male, persons (%)	47 (36.4%)	125 (49.2%)	
Female, persons (%)	82 (63.6%)	129 (50.8%)	
Total	129	254	
Age, year (persons)	52.50 ± 16.19 (129)	59.20 ± 13.03 (254)	<0.001
Mean Deviation (dB)	−0.80 ± 1.31	−6.09 ± 7.16	<0.001
Pattern Standard Deviation (dB)	2.11 ± 1.37	5.53 ± 4.22	<0.001
Refraction (Diopter)	−2.00 ± 3.08	−2.47 ± 4.09	0.104
Minimum Rim Width (μm)			
T	226.26 ± 53.96	167.25 ± 56.97	<0.001
TI	329.76 ± 72.39	210.02 ± 98.30	<0.001
NI	395.67 ± 345.50	248.98 ± 96.56	<0.001
N	324.39 ± 73.35	233.70 ± 89.18	<0.001
NS	348.46 ± 78.25	251.48 ± 97.92	<0.001
TS	308.93 ± 75.29	204.86 ± 87.65	<0.001
G	305.15 ± 59.72	213.81 ± 69.19	<0.001
Peripapillary Nerve Fiber Layer Thickness (μm)			
T	87.09 ± 21.07	69.21 ± 37.36	<0.001
TI	160.41 ± 23.36	110.52 ± 47.52	<0.001
NI	111.57 ± 25.90	87.07 ± 32.58	<0.001
N	72.98 ± 21.65	62.22 ± 23.08	<0.001
NS	120.17 ± 26.12	93.80 ± 33.94	<0.001
TS	146.29 ± 24.47	105.43 ± 41.71	<0.001
G	103.67 ± 11.85	79.96 ± 22.01	<0.001
Retina Average Thickness, ETDRS grid (μm)			
T1	324.78 ± 14.04	310.74 ± 20.48	<0.001
T2	277.56 ± 13.41	267.13 ± 18.39	<0.001
I1	334.38 ± 15.44	317.69 ± 26.93	<0.001
I2	281.76 ± 15.94	265.88 ± 22.83	<0.001
N1	339.52 ± 16.34	326.72 ± 22.37	<0.001
N2	313.23 ± 18.58	297.97 ± 24.66	<0.001
S1	338.07 ± 15.20	324.60 ± 21.97	<0.001
S2	296.18 ± 14.58	282.64 ± 20.56	<0.001
C	268.37 ± 23.14	263.10 ± 28.03	0.010
Nerve Fiber Layer, ETDRS grid (μm)			
T1	17.43 ± 1.55	18.27 ± 3.93	0.001
T2	20.05 ± 4.11	19.25 ± 4.59	0.020
I1	26.17 ± 3.22	23.46 ± 5.36	<0.001
I2	41.10 ± 6.59	31.15 ± 10.26	<0.001
N1	21.07 ± 2.61	20.76 ± 4.41	0.294
N2	49.09 ± 8.64	41.11 ± 10.89	<0.001
S1	24.37 ± 2.91	22.88 ± 5.45	<0.001
S2	39.40 ± 5.43	32.70 ± 9.43	<0.001
C	11.54 ± 2.38	10.87 ± 4.74	0.037
Ganglion Cell Layer, ETDRS grid (μm)			
T1	47.62 ± 5.28	36.76 ± 10.79	<0.001
T2	34.97 ± 4.06	27.56 ± 7.39	<0.001
I1	51.97 ± 4.22	41.85 ± 11.24	<0.001
I2	32.06 ± 3.38	27.30 ± 5.90	<0.001
N1	50.44 ± 5.03	42.16 ± 10.72	<0.001
N2	39.11 ± 3.80	34.35 ± 6.27	<0.001
S1	52.06 ± 5.29	43.71 ± 10.52	<0.001
S2	35.31 ± 3.56	30.39 ± 5.96	<0.001
C	14.39 ± 5.16	12.96 ± 5.39	<0.001
Inner Plexiform Layer, ETDRS grid (μm)			
T1	41.13 ± 4.04	35.54 ± 6.55	<0.001
T2	32.19 ± 2.45	28.48 ± 4.35	<0.001
I1	40.78 ± 2.88	35.27 ± 6.74	<0.001
I2	26.43 ± 2.68	24.11 ± 3.98	<0.001
N1	42.02 ± 3.12	38.07 ± 18.14	<0.001
N2	30.57 ± 2.81	27.84 ± 4.34	<0.001
S1	40.86 ± 3.63	36.54 ± 6.32	<0.001
S2	28.74 ± 2.90	26.15 ± 4.04	<0.001
C	19.47 ± 3.97	18.33 ± 4.53	<0.001
Retina Average Thickness, 8 × 8 grid (μm)			
1.1	0.23 ± 0.02,	0.22 ± 0.03	0.002
1.2	0.24 ± 0.01	0.23 ± 0.03	<0.001
1.3	0.25 ± 0.02	0.23 ± 0.02	<0.001
1.4	0.26 ± 0.02	0.24 ± 0.02	<0.001
1.5	0.27 ± 0.02	0.25 ± 0.03	<0.001
1.6	0.29 ± 0.02	0.26 ± 0.03	<0.001
1.7	0.30 ± 0.02	0.26 ± 0.04	<0.001
1.8	0.28 ± 0.03	0.25 ± 0.07	<0.001
2.1	0.23 ± 0.01	0.22 ± 0.02	<0.001
2.2	0.24 ± 0.01	0.23 ± 0.02	<0.001
2.3	0.26 ± 0.01	0.25 ± 0.02	<0.001
2.4	0.28 ± 0.02	0.26 ± 0.02	<0.001
2.5	0.29 ± 0.02	0.27 ± 0.02	<0.001
2.6	0.29 ± 0.02	0.27 ± 0.02	<0.001
2.7	0.30 ± 0.02	0.27 ± 0.03	<0.001
2.8	0.32 ± 0.02	0.28 ± 0.05	<0.001
3.1	0.24 ± 0.01	0.23 ± 0.02	<0.001
3.2	0.27 ± 0.01	0.25 ± 0.02	<0.001
3.3	0.30 ± 0.02	0.28 ± 0.03	<0.001
3.4	0.33 ± 0.02	0.31 ± 0.03	<0.001
3.5	0.33 ± 0.02	0.32 ± 0.03	<0.001
3.6	0.32 ± 0.02	0.30 ± 0.02	<0.001
3.7	0.31 ± 0.02	0.29 ± 0.03	<0.001
3.8	0.32 ± 0.03	0.28 ± 0.04	<0.001
4.1	0.25 ± 0.01	0.24 ± 0.03	<0.001
4.2	0.28 ± 0.01	0.27 ± 0.03	<0.001
4.3	0.32 ± 0.02	0.31 ± 0.03	<0.001
4.4	0.31 ± 0.02	0.30 ± 0.03	<0.001
4.5	0.31 ± 0.02	0.30 ± 0.03	<0.001
4.6	0.34 ± 0.02	0.33 ± 0.02	<0.001
4.7	0.33 ± 0.02	0.30 ± 0.02	<0.001
4.8	0.30 ± 0.02	0.28 ± 0.04	<0.001
5.1	0.25 ± 0.01	0.24 ± 0.02	<0.001
5.2	0.28 ± 0.01	0.27 ± 0.02	<0.001
5.3	0.32 ± 0.02	0.30 ± 0.03	<0.001
5.4	0.31 ± 0.02	0.30 ± 0.02	<0.001
5.5	0.31 ± 0.02	0.30 ± 0.03	<0.001
5.6	0.34 ± 0.02	0.33 ± 0.02	<0.001
5.7	0.32 ± 0.02	0.31 ± 0.02	<0.001
5.8	0.30 ± 0.02	0.28 ± 0.04	<0.001
6.1	0.24 ± 0.01	0.24 ± 0.02	<0.001
6.2	0.27 ± 0.01	0.26 ± 0.02	<0.001
6.3	0.30 ± 0.02	0.29 ± 0.02	<0.001
6.4	0.33 ± 0.02	0.32 ± 0.02	<0.001
6.5	0.34 ± 0.02	0.32 ± 0.02	<0.001
6.6	0.33 ± 0.02	0.31 ± 0.02	<0.001
6.7	0.31 ± 0.02	0.30 ± 0.02	<0.001
6.8	0.31 ± 0.03	0.29 ± 0.03	<0.001
7.1	0.23 ± 0.01	0.23 ± 0.02	0.001
7.2	0.25 ± 0.01	0.24 ± 0.02	<0.001
7.3	0.27 ± 0.01	0.26 ± 0.02	<0.001
7.4	0.29 ± 0.02	0.28 ± 0.02	<0.001
7.5	0.30 ± 0.02	0.29 ± 0.02	<0.001
7.6	0.30 ± 0.02	0.28 ± 0.02	<0.001
7.7	0.30 ± 0.02	0.28 ± 0.02	<0.001
7.8	0.31 ± 0.02	0.28 ± 0.03	<0.001
8.1	0.23 ± 0.01	0.22 ± 0.03	<0.001
8.2	0.24 ± 0.01	0.23 ± 0.02	<0.001
8.3	0.25 ± 0.01	0.24 ± 0.02	<0.001
8.4	0.26 ± 0.01	0.25 ± 0.02	<0.001
8.5	0.27 ± 0.02	0.26 ± 0.02	<0.001
8.6	0.28 ± 0.02	0.26 ± 0.02	<0.001
8.7	0.29 ± 0.02	0.27 ± 0.03	<0.001
8.8	0.31 ± 0.02	0.28 ± 0.03	<0.001

T: temporal; TI: temporal inferior; TS: temporal superior; N: nasal; NI: nasal inferior; NS: nasal superior; G: global; T1: inner temporal; T2: outer temporal; I1: inner inferior; I2: outer inferior; N1: inner nasal; N2: outer nasal; S1: inner superior; S2: outer superior; C: central. * Two-sided independent *t*-test.

**Table 3 diagnostics-12-00391-t003:** Optical coherence tomography (OCT)-related feature groups, as well as the support vector machine (SVM) classification results.

		10-Fold Cross-Validation			
Normal		254 eyes			
Glaucoma		498 eyes			
Feature Group	Amounts of Features	Sensitivity	Specificity	Accuracy	AUC
OCT	114	0.85	0.70	0.80	0.82
MRW	7	0.81	0.70	0.77	0.81
ppNFLT	7	0.81	0.71	0.77	0.81
RAT	9	0.82	0.46	0.70	0.72
NFL	9	0.80	0.62	0.74	0.79
GCL	9	0.78	0.68	0.75	0.80
IPL	9	0.76	0.68	0.74	0.78
RAT 8 × 8	64	0.84	0.59	0.76	0.80

OCT: all optical coherence tomography parameters; MRW: minimal rim width; ppNFLT: peripapillary nerve fiber layer thickness; RAT: retinal average thickness in 1, 3, 6 mm ETDRS grid; NFL: nerve fiber layer in 1, 3, 6 mm ETDRS grid; GCL: ganglion cell layer in 1, 3, 6 mm ETDRS grid; IPL: inner plexiform layer in 1, 3, 6 mm ETDRS grid; RAT: retinal average thickness in 8 × 8 grid; AUC: area under the receiver operating characteristic curve.

**Table 4 diagnostics-12-00391-t004:** OCT-related feature groups (plus age, gender, and refraction) and the SVM classification results.

		10-Fold Cross-Validation			
Normal		254 eyes			
Glaucoma		498 eyes			
Feature Group	Amounts of Features	Sensitivity	Specificity	Accuracy	AUC
AGR	3	0.97	0.12	0.69	0.55
OCT + AGR	117	0.85	0.67	0.79	0.82
MRW + AGR	10	0.82	0.67	0.75	0.81
ppNFLT + AGR	10	0.84	0.65	0.78	0.82
RAT + AGR	12	0.87	0.40	0.71	0.71
NFL + AGR	12	0.87	0.56	0.76	0.78
GCL + AGR	12	0.84	0.63	0.77	0.79
IPL + AGR	12	0.82	0.58	0.74	0.76
RAT 8 × 8 + AGR	67	0.85	0.55	0.75	0.58

AGR: age, gender, refraction; OCT: all optical coherence tomography parameters; MRW: minimal rim width; ppNFLT: peripapillary nerve fiber layer thickness; RAT: retinal average thickness in 1, 3, 6 mm ETDRS grid; NFL: nerve fiber layer in 1, 3, 6 mm ETDRS grid; GCL: ganglion cell layer in 1, 3, 6 mm ETDRS grid; IPL: inner plexiform layer in 1, 3, 6 mm ETDRS grid; RAT: retinal average thickness in 8 × 8 grid; AUC: area under the receiver operating characteristic curve.

**Table 5 diagnostics-12-00391-t005:** SVM classification results between normal and different stage of glaucomatous eyes, using each OCT-related feature groups.

	10-Fold Cross-Validation			
Normal	254 eyes			
Early stage	337 eyes			
Feature Group	Sensitivity	Specificity	Accuracy	AUC
OCT	0.87	0.72	0.75	0.78
MRW	0.71	0.74	0.73	0.78
ppNFLT	0.70	0.76	0.73	0.77
RAT	0.81	0.69	0.76	0.68
NFL	0.70	0.68	0.73	0.73
GCL	0.74	0.79	0.76	0.77
IPL	0.66	0.77	0.70	0.74
RAT 8 × 8	0.72	0.66	0.72	0.76
Normal	254 eyes			
Moderate stage	73 eyes			
Feature Group	Sensitivity	Specificity	Accuracy	AUC
OCT	0.72	0.96	0.91	0.89
MRW	0.60	0.97	0.89	0.86
ppNFLT	0.74	0.97	0.92	0.87
RAT	0.54	0.98	0.88	0.78
NFL	0.70	0.98	0.89	0.86
GCL	0.72	0.98	0.92	0.85
IPL	0.64	0.97	0.90	0.86
RAT 8 × 8	0.64	0.96	0.89	0.86
Normal	254 eyes			
Severe stage	88 eyes			
Feature Group	Sensitivity	Specificity	Accuracy	AUC
OCT	0.96	0.96	0.96	0.93
MRW	0.94	0.94	0.94	0.91
ppNFLT	0.96	0.96	0.96	0.90
RAT	0.93	0.93	0.93	0.90
NFL	0.95	0.95	0.95	0.92
GCL	0.95	0.95	0.95	0.92
IPL	0.94	0.94	0.94	0.90
RAT 8 × 8	0.95	0.95	0.95	0.93

OCT: all optical coherence tomography parameters; MRW: minimal rim width; ppNFLT: peripapillary nerve fiber layer thickness; RAT: retinal average thickness in 1, 3, 6 mm ETDRS grid; NFL: nerve fiber layer in 1, 3, 6 mm ETDRS grid; GCL: ganglion cell layer in 1, 3, 6 mm ETDRS grid; IPL: inner plexiform layer in 1, 3, 6 mm ETDRS grid; RAT: retinal average thickness in 8 × 8 grid; AUC: area under the receiver operating characteristic curve.

**Table 6 diagnostics-12-00391-t006:** SVM classification results between normal and different stage of glaucomatous eyes using each OCT-related feature groups (plus age, gender, and refraction).

	10-Fold Cross-Validation			
Normal	254 eyes			
Early stage	337 eyes			
Feature Group	Sensitivity	Specificity	Accuracy	AUC
AGR	0.91	0.18	0.60	0.58
OCT + AGR	0.79	0.71	0.75	0.78
MRW + AGR	0.73	0.70	0.72	0.77
ppNFLT + AGR	0.76	0.72	0.74	0.77
RAT + AGR	0.74	0.47	0.72	0.67
NFL + AGR	0.80	0.64	0.73	0.73
GCL + AGR	0.73	0.76	0.70	0.75
IPL + AGR	0.73	0.76	0.70	0.72
RAT 8 × 8 + AGR	0.76	0.60	0.69	0.58
Normal	254 eyes			
Moderate stage	73 eyes			
Feature Group	Sensitivity	Specificity	Accuracy	AUC
AGR	0.04	1	0.77	0.66
OCT + AGR	0.72	0.96	0.91	0.89
MRW + AGR	0.60	0.97	0.89	0.86
ppNFLT + AGR	0.78	0.97	0.93	0.89
RAT + AGR	0.45	0.97	0.85	0.79
NFL + AGR	0.67	0.97	0.91	0.84
GCL + AGR	0.72	0.97	0.91	0.85
IPL + AGR	0.63	0.97	0.89	0.84
RAT 8 × 8 + AGR	0.63	0.95	0.88	0.75
Normal	254 eyes			
Severe stage	88 eyes			
Feature Group	Sensitivity	Specificity	Accuracy	AUC
AGR	0.73	0.73	0.73	0.68
OCT + AGR	0.96	0.96	0.96	0.93
MRW + AGR	0.94	0.94	0.94	0.92
ppNFLT + AGR	0.96	0.96	0.96	0.92
RAT + AGR	0.92	0.92	0.92	0.88
NFL + AGR	0.95	0.95	0.95	0.93
GCL + AGR	0.95	0.95	0.95	0.92
IPL + AGR	0.94	0.94	0.94	0.90
RAT 8 × 8 + AGR	0.68	0.68	0.73	0.70

AGR: age, gender, refraction; OCT: all optical coherence tomography parameters; MRW: minimal rim width; ppNFLT: peripapillary nerve fiber layer thickness; RAT: retinal average thickness in 1, 3, 6 mm ETDRS grid; NFL: nerve fiber layer in 1, 3, 6 mm ETDRS grid; GCL: ganglion cell layer in 1, 3, 6 mm ETDRS grid; IPL: inner plexiform layer in 1, 3, 6 mm ETDRS grid; RAT: retinal average thickness in 8 × 8 grid; AUC: area under the receiver operating characteristic curve.

**Table 7 diagnostics-12-00391-t007:** Selected feature subsets with mutual information method.

Name	Number of Features	Feature Subset
MI 1	1	MRW_TI
MI 2	2	MRW_TI, MRW_G
MI 4	4	MRW_TI, MRW_G, GCL_T2, GCL_I1
MI 6	6	MRW_TI, MRW_G, GCL_T2, GCL_I1, ppNFLT_TS, GCL_T1
MI 8	8	MRW_TI, MRW_G, GCL_T2, GCL_I1, ppNFLT_TS, GCL_T1, ppNFLT_TI, IPL_T2
MI 10	10	MRW_TI, MRW_G, GCL_T2, GCL_I1, ppNFLT_TS, GCL_T1, ppNFLT_TI, IPL_T2, MRW_TS, ppNFLT_G

MI: mutual information; MRW: minimal rim width; GCL: ganglion cell layer; ppNFLT: peripapillary nerve fiber layer thickness; TI: temporal inferior; TS: temporal superior; G: global; T1: inner temporal; T2: outer temporal; I1: inner inferior.

**Table 8 diagnostics-12-00391-t008:** SVM classification results using selected feature subsets between normal and glaucomatous eyes.

	10-Fold Cross-Validation			
Name	Sensitivity	Specificity	Accuracy	AUC
MI 1	0.84	0.60	0.76	0.80
MI 2	0.83	0.70	0.78	0.81
MI 4	0.83	0.70	0.79	0.82
MI 6	0.86	0.70	0.80	0.84
MI 8	0.85	0.69	0.80	0.84
MI 10	0.85	0.71	0.80	0.83

MI: mutual information; AUC: area under the receiver operating characteristic curve.

## Data Availability

Not applicable.

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
