# Peer review of "Glaucoma Detection Using Support Vector Machine Method Based on Spectralis OCT"

_diagnostics, 2022, doi:10.3390/diagnostics12020391_

Round 1
Reviewer 1 Report
The authors presented a very interesting study analyzing the diagnostic capability of parameters derived from Spectralis OCT parameters on glaucoma detection using Support Vector Machine classification method. The manuscript is absolutely with merit and the findings are worth reporting. However, before publication could be considered, the authors should revise the manuscript and address the following comments:
ABSTRACT
- The abbreviations should be explained the first time that they are used (i.e. line 17: the explanation “optical coherence tomography” should be provided with the abbreviation “OCT”; the same for “AUC” - area under the curve - at line 24)
METHODS
- The authors use the term “glaucoma”: can you please specify what type of glaucoma was considered (primary/secondary; if secondary which type; open-angle/angle closure glaucoma, normal tension glaucoma)?
- Statistics: The author should provide a statistical power estimation for their study or at least some justification of the study n and add it to the methods
RESULTS
- The authors state that there was a statistically significant difference in age and between the two groups - it may be interesting to adjust the results for age
DISCUSSION
- The authors should discuss the fact that the healthy and glaucoma group were different specifically for age and add this as limitation to the “Limitations section” of the discussion
- The authors should expand the discussion providing some insight on the application and use of Support Vector Machine classification for glaucoma in the clinical practice in ophthalmology and its future directions of application
FIGURE/TABLES
Please revise the content and legends and provide the complete explanation of the abbreviations used.
Author Response
Response to the Reviewer one’s comment
The authors presented a very interesting study analyzing the diagnostic capability of parameters derived from Spectralis OCT parameters on glaucoma detection using Support Vector Machine classification method. The manuscript is absolutely with merit and the findings are worth reporting. However, before publication could be considered, the authors should revise the manuscript and address the following comments:
- Reply: Thanks so much for your support for our study. We have tried our best to address each of your concern and suggestion and have revised our work as your precious opinions.
ABSTRACT
- The abbreviations should be explained the first time that they are used (i.e. line 17: the explanation “optical coherence tomography” should be provided with the abbreviation “OCT”; the same for “AUC” - area under the curve - at line 24)
- Reply: Thank you so much for your kind reminding. We have provided the abbreviation of OCT and AUC in the abstract part.
METHODS
- The authors use the term “glaucoma”: can you please specify what type of glaucoma was considered (primary/secondary; if secondary which type; open-angle/angle closure glaucoma, normal tension glaucoma)?
- Reply: Thank you for much for your question. Actually, in this kind of imaging study, glaucoma subjects were recruited from primary glaucoma including open angle and angle closure cases. We have added more description in the Participants part (please see line 64, 76).
- Statistics: The author should provide a statistical power estimation for their study or at least some justification of the study n and add it to the methods
- Reply: Thanks so much for your concern. Our study was case-control design but not sampling/ experiment design. Inclusion and exclusion criteria should be based on several clinical information, including refraction error, glaucoma severity from visual field data and imaging quality. Therefore, G power tool is not appropriate in our study.
RESULTS
- The authors state that there was a statistically significant difference in age and between the two groups - it may be interesting to adjust the results for age
- Reply: Thank you so much for your nice opinion. Actually, we did adjust the results for age in the Table 6 in the first version of manuscript. Besides of age, gender and refraction will influence RNFL thickness and other parameters as well; therefore, the three factors including age, gender, refraction should be adjusted at the same time.
DISCUSSION
- The authors should discuss the fact that the healthy and glaucoma group were different specifically for age and add this as limitation to the “Limitations section” of the discussion
- Reply: Thank you so much for your great comment. We have added this point in the discussion part. ( please see: Line 339-343) 
- The authors should expand the discussion providing some insight on the application and use of Support Vector Machine classification for glaucoma in the clinical practice in ophthalmology and its future directions of application
- Reply: Thanks so much for your great comment. We have added some discussion about the SVM classification in glaucoma practice in ophthalmology. (please see Line 296-303)
FIGURE/TABLES
Please revise the content and legends and provide the complete explanation of the abbreviations used.
- Reply: Thank you so much for your reminding and we have revised the whole contents and legends and provided the complete explanation of the abbreviations used in the Tables/ Figure.
Reviewer 2 Report
The paper needs serious improvements in order to be at a good level for publishing.
The following aspects must be improved:
- a section of related works must be added (other methods that perform the same processing must be compared with advantages and disadvantages)
- what is the novelty of the paper
- based on what reasons the SVM was used for the proposed method
- what kernels were used
- comparison with other existing methods must be added
- figures S1 and S2 must be correctly numbered
Author Response
Response to the Reviewers Two’s comments:
The paper needs serious improvements in order to be at a good level for publishing.
- Reply: thank so much for your great support and encouragement for our research .
The following aspects must be improved:
- a section of related works must be added (other methods that perform the same processing must be compared with advantages and disadvantages)
- what is the novelty of the paper
- based on what reasons the SVM was used for the proposed method
- what kernels were used
- comparison with other existing methods must be added
- figures S1 and S2 must be correctly numbered
Reply: We have followed your suggestions. We have added more information about the novelty of our paper and advantages of this proposed SVM model. We also compared our results with other previous studies in the discussion part. (please see line 268-303) The figures S1 and S2 have been correctly
Reviewer 3 Report
This study investigated the diagnostic capability of Spectralis OCT parameters on glaucoma detection using the support vector machine (SVM) classification method in the Asian population. A total of 498 glaucomatous eyes and 254 normal eyes were included. The authors found that applying all OCT features with the SVM method had good capability in the detection of glaucomatous eyes. The authors also found that minimum rim width (MRW) is a good feature group to discriminate early glaucomatous from normal eyes.
Please list detailed methods for Spectralis OCT imaging, such as parameters measured. A representative image will be helpful.
Author Response
Response to the Reviewer Three’s comments:
- This study investigated the diagnostic capability of Spectralis OCT parameters on glaucoma detection using the support vector machine (SVM) classification method in the Asian population. A total of 498 glaucomatous eyes and 254 normal eyes were included. The authors found that applying all OCT features with the SVM method had good capability in the detection of glaucomatous eyes. The authors also found that minimum rim width (MRW) is a good feature group to discriminate early glaucomatous from normal eyes.
- Please list detailed methods for Spectralis OCT imaging, such as parameters measured. A representative image will be helpful.
- Thank you so much for your great support and encouragement for our study. We have already listed detailed information of parameters of Spectralis OCT in the Table 1 and 2. We have provided a representative image (Figure 1) in the revised text.  
Round 2
Reviewer 1 Report
The authors addressed the comments in a detailed way.
The only comment left is that there are still some abbreviations in the abstract that should be explained before publication - the reader should be able to completely understand the abstract without the need to refer to the main document (e.g. line 28-30: please provide the explanation for TI, G, T2, I1, TS, T1)
Author Response
The authors addressed the comments in a detailed way.
The only comment left is that there are still some abbreviations in the abstract that should be explained before publication - the reader should be able to completely understand the abstract without the need to refer to the main document (e.g. line 28-30: please provide the explanation for TI, G, T2, I1, TS, T1)
- Reply: Thanks so much for your great comment. We have followed your suggestions to do some changes in this part.
Reviewer 2 Report
I recommend to add a section related work in which to include all existing methods (together with their results) that are introduces in section Discussion. A comparison between them must be done.
The explanation of choosing the SVM must be made more clearly.: what is the advantage of using SVM?
Comparison with other existing solutions (based on results) must be added.
Author Response
The paper needs serious improvements in order to be at a good level for publishing.
The following aspects must be improved:
I recommend to add a section related work in which to include all existing methods (together with their results) that are introduces in section Discussion. A comparison between them must be done.
- Reply: Thanks so much for your opinion. We have added a section to compare our results with other previous studies using Spectralis OCT in the discussion section ( please see: lines 315-342) .
“To our knowledge, our study was the few ones which evaluated the application of machine learning technique in complicated Spectralis OCT parameters for glaucoma detection including ppRNFL, ONH and macular parameters. Several published literatures have explored the use of Spectralis OCT parameters to construct machine learning classifiers for glaucoma diagnosis [16,37-39]. Kim et al. developed several machine learning models including SVM for glaucoma diagnosis using ppRNFL parameters plus clinical features (age, IOP, and corneal thickness) and visual field information, and they found the random forest model had the best performance with AUC value of 0.979, and the AUC value of the SVM model is 0.967 [37]. Oh et al. also constructed several machine learning models including SVM using 3 of ppRNFL measurements (ppRNFL superior, ppRNFL inferior, and ppRNFL temporal) plus IOP and PSD for glaucoma detection, and the extreme gradient boosting model was shown to be the best model with AUC value of 0.945, the same AUC value as the SVM’s but with higher accuracy, sensitivity and specificity. ppRNFL superior, ppRNFL inferior and PSD were found to have stronger influence in their proposed prediction model [16]. Park et al. used a multilayer neural network to combine BMO-MRW and ppRNFL parameters for glaucoma diagnosis, which showed better performance than using either BMO-MRW or ppRNFL data alone [38]. Deep learning classification model was adopted by Seo et al. for discriminating early normal tension glaucoma from glaucoma suspect and showed best performance considering 3 OCT-based parameters together (BMO-MRW, ppRNFL, and the color classification of ppRNFL) with AUC value of 0.966 [39]. Though it is difficult to directly compare our results with previous research due to the differences in the subjects included and the OCT parameters and machine learning methods used, the above papers and ours had proved that it is feasible to construct reliable machine learning classifiers using Spectralis OCT parameters for glaucoma diagnosis. Unlike previous studies, our study not only used the ONH and the ppRNFL parameters but also covered macula-related parameters in order to have a more comprehensive analysis. “
The explanation of choosing the SVM must be made more clearly.: what is the advantage of using SVM?
- Reply: Thanks so much again for your opinion. We have explained the rationale of choosing SVM in this study more clearly and we also added the advantages of SVM in the revised text (Please see lines: 277-288).
“ SVM is a supervised machine learning classifier, which is one of the most powerful and robust classification and widely used to deal with binary classification problems in various fields [24,30-33]. It has also been used for glaucoma detection in previous studies and provided promising results [19,34-36]. Compared with other machine learning approaches, SVM maps the nonlinearly separable data into a high-dimensional space through kernel functions to transfer the corresponding to a linearly separable state. It maintains high generalization ability of the learning machine simultaneously. Thus, SVM is relatively effective when solving problems with the number of feature dimensions greater than the number of samples [31], as in this study we used abundant OCT parameters for glaucoma discrimination. In addition, for small data problems like ours, SVM still performs well in accuracy and is relatively memory efficient [24,33]. “
Comparison with other existing solutions (based on results) must be added.
- Reply: Thanks so much for your suggestion. We have added a section to compare our results with other previous studies using Spectralis OCT in the discussion section ( please see: lines 315-342) .
“To our knowledge, our study was the few ones which evaluated the application of machine learning technique in complicated Spectralis OCT parameters for glaucoma detection including ppRNFL, ONH and macular parameters. Several published literatures have explored the use of Spectralis OCT parameters to construct machine learning classifiers for glaucoma diagnosis [16,37-39]. Kim et al. developed several machine learning models including SVM for glaucoma diagnosis using ppRNFL parameters plus clinical features (age, IOP, and corneal thickness) and visual field information, and they found the random forest model had the best performance with AUC value of 0.979, and the AUC value of the SVM model is 0.967 [37]. Oh et al. also constructed several machine learning models including SVM using 3 of ppRNFL measurements (ppRNFL superior, ppRNFL inferior, and ppRNFL temporal) plus IOP and PSD for glaucoma detection, and the extreme gradient boosting model was shown to be the best model with AUC value of 0.945, the same AUC value as the SVM’s but with higher accuracy, sensitivity and specificity. ppRNFL superior, ppRNFL inferior and PSD were found to have stronger influence in their proposed prediction model [16]. Park et al. used a multilayer neural network to combine BMO-MRW and ppRNFL parameters for glaucoma diagnosis, which showed better performance than using either BMO-MRW or ppRNFL data alone [38]. Deep learning classification model was adopted by Seo et al. for discriminating early normal tension glaucoma from glaucoma suspect and showed best performance considering 3 OCT-based parameters together (BMO-MRW, ppRNFL, and the color classification of ppRNFL) with AUC value of 0.966 [39]. Though it is difficult to directly compare our results with previous research due to the differences in the subjects included and the OCT parameters and machine learning methods used, the above papers and ours had proved that it is feasible to construct reliable machine learning classifiers using Spectralis OCT parameters for glaucoma diagnosis. Unlike previous studies, our study not only used the ONH and the ppRNFL parameters but also covered macula-related parameters in order to have a more comprehensive analysis. “
Round 3
Reviewer 2 Report
Some of my comments are still not addressed:
- A section of related work must be added (not included in the Discussion section). This section must contain other existing methods with their results and a comparison between them.
- Comparison with other existing solutions (based on results) must be added - data used in the current paper must be analysed with other methods and a comparison of the results must be done.
- Reply: Thank you so much for your great support about our research work. Regarding my manuscript, Here we would like to discuss with you about the reviewer’s comments.
At first time revision, we were asked to revise our work based on the three reviewers’ comments. And we did our best to make changes according to the reviewers’ suggestions and opinions. The reviewer 3 has already accepted our changes after first time revision. Through the second time of revision, the reviewer 1 has also accepted our revision. Regarding the reviewer two’ comments, he insisted that we should use our current database to do more analysis and compared our current study results with the other existing machine learning (ML) methods, which is not really adequate. Actually, we have already tried our best to address more in the discussion part to compare our current study with other previous published work related to Spectralis OCT (please see the reply letter to the reviewer as shown below). Furthermore, since there are so many ML methods, it is not reasonable to ask us to do more
analysis using so many ML methods and compared all of them, which is also not our study purpose. We feel very unfair in the review process regarding the inadequate request made by one the three reviewers. We would like to emphasize again that our study purpose to is to focus on SVM methodology in glaucoma diagnosis but not comparisons with other existing methods.
Therefore, we sincerely request you to make more neutral judgment and decision about our manuscript based on your expertise. Your kind understanding is highly appreciated.